# CONTINUOUS TRANSFER LEARNING

## ABSTRACT

Transfer learning has been successfully applied across many high-impact applications. However, most existing work focuses on the static transfer learning setting, and very little is devoted to modeling the time evolving target domain, such as the online reviews for movies. To bridge this gap, in this paper, we focus on the continuous transfer learning setting with a time evolving target domain. One major challenge associated with continuous transfer learning is the time evolving relatedness of the source domain and the current target domain as the target domain evolves over time. To address this challenge, we first derive a generic generalization error bound on the current target domain with flexible domain discrepancy measures. Furthermore, a novel label-informed $\mathcal{C}$-divergence is proposed to measure the shift of joint data distributions (over input features and output labels) across domains. It could be utilized to instantiate a tighter error upper bound in the continuous transfer learning setting, thus motivating us to develop an adversarial Variational Auto-encoder algorithm named CONTE by minimizing the $\mathcal{C}$-divergence based error upper bound. Extensive experiments on various data sets demonstrate the effectiveness of our CONTE algorithm.

## 1 INTRODUCTION

Transfer learning has achieved significant success across multiple high-impact application domains (Pan & Yang, 2009). Compared to conventional machine learning methods assuming both training and test data have the same data distribution, transfer learning allows us to learn the target domain with limited label information by leveraging a related source domain with abundant label information (Ying et al., 2018). However, in many real applications, the target domain is constantly evolving over time.

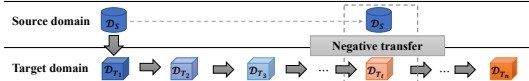

Figure 1: Illustration of continuous transfer learning. It learns a predictive function in $\mathcal{D}_{T_t}$ using knowledge from both source domain $\mathcal{D}_S$ and historical target domain $\mathcal{D}_{T_i}(i = 1, \cdots, t-1)$. Directly transferring from the source domain $\mathcal{D}_S$ to the target domain $\mathcal{D}_{T_t}$ might lead to negative transfer with undesirable predictive performance.

For example, the online movie reviews are changing over the years: some famous movies were not well received by the mainstream audience when they were first released, but became famous only years later (e.g., *Citizen Cane*, *Fight Club*, and *The Shawshank Redemption*); whereas the online book reviews typically do not have this type of dynamics. It is challenging to transfer knowledge from the static source domain (e.g., the book reviews) to the time evolving target domain (e.g., the movie reviews). Therefore, in this paper, we study the transfer learning setting with a static source domain and a continuously time evolving target domain (see Figure 1), which has not attracted much attention from the research community and yet is commonly seen across many real applications. The unique challenge for continuous transfer learning lies in the time evolving nature of the task relatedness between the static source domain and the time evolving target domain. Although the change in the target data distribution in consecutive time stamps might be small, over time, the cumulative change in the target domain might even lead to negative transfer (Rosenstein et al., 2005).

Existing theoretical analysis on transfer learning (Ben-David et al., 2010; Mansour et al., 2009) showed that the target error is typically bounded by the source error, the domain discrepancy of marginal data distributions and the difference of labeling functions. However, it has been observed (Zhao et al., 2019; Wu et al., 2019) that marginal feature distribution alignment might not guarantee the minimization of the target error in real world scenarios. This indicates that in the context of continuous transfer learning, marginal feature distribution alignment would lead to the sub-optimal solution (or even negative transfer) with undesirable predictive performance when directly transferring from $\mathcal{D}_S$ to the target domain $\mathcal{D}_{T_t}$ at the $t^{\text{th}}$ time stamp. This paper aims to bridge the gap in terms of both the theoretical analysis and the empirical solutions for the target domain with a time evolving distribution, which lead to a novel continuous transfer learning algorithm as

well as the characterization of negative transfer. The main contributions of this paper are summarized as follows: (1) We derive a generic error bound for continuous transfer learning setting with flexible domain divergence measures; (2) We propose a label-informed domain discrepancy measure ($\mathcal{C}$-divergence) with its empirical estimate, which instantiates a tighter error bound for continuous transfer learning setting; (3) Based on the proposed $\mathcal{C}$-divergence, we design a novel adversarial Variational Auto-encoder algorithm (CONTE) for continuous transfer learning; (4) Extensive experimental results on various data sets verify the effectiveness of the proposed CONTE algorithm.

The rest of the paper is organized as follows. Section 2 introduces the notation and our problem definition. We derive a generic error bound for continuous transfer learning setting in Section 3. Then we propose a novel $\mathcal{C}$-divergence in Section 4, followed by a instantiated error bound and a novel continuous transfer learning algorithm in Section 5. The experimental results are provided in Section 6. We summarize the related work in Section 7, and conclude the paper in Section 8.

## 2 PRELIMINARIES

In this section, we introduce the notation and problem definition of continuous transfer learning.

### 2.1 NOTATION

We use $\mathcal{X}$ and $\mathcal{Y}$ to denote the input space and label space. Let $\mathcal{D}_S$ and $\mathcal{D}_T$ denote the source and target domains with data distribution $p_S(\mathbf{x}, y)$ and $p_T(\mathbf{x}, y)$ over $\mathcal{X} \times \mathcal{Y}$, respectively. Let $\mathcal{H}$ be a hypothesis class on $\mathcal{X}$, where a hypothesis is a function $h : \mathcal{X} \to \mathcal{Y}$. The notation is summarized in Table 3 in the appendices.

### 2.2 PROBLEM DEFINITION

Transfer learning (Pan & Yang, 2009) refers to the knowledge transfer from source domain to target domain such that the prediction performance on the target domain could be significantly improved as compared to learning from the target domain alone. However, in some applications, the target domain is changing over time, hence the time evolving relatedness between the source and target domains. This motivates us to consider the transfer learning setting with the time evolving target domain, which is much less studied as compared to the static transfer learning setting. We formally define the continuous transfer learning problem as follows.

**Definition 2.1.** (*Continuous Transfer Learning*) Given a source domain $\mathcal{D}_S$ (available at time stamp $j = 1$) and a time evolving target domain $\{\mathcal{D}_{T_j}\}_{j=1}^n$ with time stamp $j$, *continuous transfer learning* aims to improve the prediction function for target domain $\mathcal{D}_{T_{t+1}}$ using the knowledge from source domain $\mathcal{D}_S$ and the historical target domain $\mathcal{D}_{T_j}(j = 1, \cdots, t)$.

Notice that the source domain $\mathcal{D}_S$ can be considered a special initial domain for the time-evolving target domain. Therefore, for notation simplicity, we will use $\mathcal{D}_{T_0}$ to represent the source domain in this paper. It assumes that there are $m_{T_0}$ labeled source examples drawn independently from a source domain $\mathcal{D}_{T_0}$ and $m_{T_j}$ labeled target examples drawn independently from a target domain $\mathcal{D}_{T_j}$ at time stamp $j$.

## 3 A GENERIC ERROR BOUND

Given a static source domain and a time evolving target domain, continuous transfer learning aims to improve the target predictive function over $\mathcal{D}_{T_{t+1}}$ using the source domain and historical target domain. We begin by considering the binary classification setting, i.e., $\mathcal{Y} = \{0, 1\}$. The source error of a hypothesis $h$ can be defined as follows: $\epsilon_{T_0}(h) = \mathbb{E}_{(\mathbf{x},y) \sim p_{T_0}(\mathbf{x},y)}\left[\mathcal{L}(h(\mathbf{x}), y)\right]$ where $\mathcal{L}(\cdot, \cdot)$ is the loss function. Its empirical estimate using source labeled examples is denoted as $\hat{\epsilon}_{T_0}(h)$. Similarly, we define the target error $\epsilon_{T_j}(h)$ and the empirical estimate of the target error $\hat{\epsilon}_{T_j}(h)$ over the target distribution $p_{T_j}(\mathbf{x}, y)$ at time stamp $j$. A natural domain discrepancy measure over joint distributions on $\mathcal{X} \times \mathcal{Y}$ between features and class labels can be defined as follows:

$$d_1(\mathcal{D}_{T_0}, \mathcal{D}_T) = \sup_{Q \in \mathcal{Q}} \left| \Pr_{\mathcal{D}_{T_0}}[Q] - \Pr_{\mathcal{D}_T}[Q] \right| \tag{1}$$

where $\mathcal{Q}$ is the set of measurable subsets under $p_{T_0}(\mathbf{x}, y)$ and $p_T(\mathbf{x}, y)$[1]. Then, the error bound of continuous transfer learning is given by the following theorem.

**Theorem 3.1.** *Assume the loss function $\mathcal{L}$ is bounded with $0 \le \mathcal{L} \le M$. Given a source domain $\mathcal{D}_{T_0}$ and historical target domain $\{\mathcal{D}_{T_i}\}_{i=1}^t$, for $h \in \mathcal{H}$, the target domain error $\epsilon_{T_{t+1}}$ on $\mathcal{D}_{t+1}$ is*

---

[1]Note that it is slightly different from $L_1$ or variation divergence in (Ben-David et al., 2010) with only marginal distribution of features involved.

*bounded as follows.*

$$\epsilon_{T_{t+1}}(h) \leq \frac{1}{\bar{\mu}} \left( \sum_{j=0}^{t} \mu^{t-j} \epsilon_{T_j}(h) + M \sum_{j=0}^{t} \mu^{t-j} d_1(\mathcal{D}_{T_j}, \mathcal{D}_{T_{t+1}}) \right)$$

*where $\mu \geq 0$ is the domain decay rate[2] indicating the importance of source or historical target domain over $\mathcal{D}_{T_{t+1}}$, and $\bar{\mu} = \sum_{j=0}^{t} \mu^{t-j}$.*

**Remark.** *In particular, we have the following arguments. (1) It is not tractable to accurately estimate $d_1$ from finite examples in real scenarios (Ben-David et al., 2010); (2) This error bound could be much tighter when considering other advanced domain discrepancy measures, e.g., $\mathcal{A}$-distance (Ben-David et al., 2007), discrepancy distance (Mansour et al., 2009), etc. (3) There are two special cases: when $\mu = 0$, the error bound of $\mathcal{D}_{T_{t+1}}$ would be simply determined by the latest historical target data $\mathcal{D}_{T_t}$, and if $\mu$ goes to infinity, $\mathcal{D}_{T_{t+1}}$ is just determined by the source data $\mathcal{D}_{T_0}$ because intuitively the coefficient $\mu^{t-j}/\bar{\mu}$ of historical target domain data $\mathcal{D}_{T_j}(j = 1, \cdots, t)$ converges to zero.*

**Corollary 3.2.** *With the assumption in Theorem 3.1 and assume that the loss function $\mathcal{L}$ is symmetric (i.e., $\mathcal{L}(y_1, y_2) = \mathcal{L}(y_2, y_1)$ for $y_1, y_2 \in \mathcal{Y}$) and obeys the triangle inequality, Then*

*(1) if $\mathcal{A}$-distance (Ben-David et al., 2007) is adopted to measure the distribution shift, i.e., $d_{\mathcal{H}\Delta\mathcal{H}} = \sup_{h,h' \in \mathcal{H}} \left| \Pr_{\mathcal{D}_{T_0}}[h(\mathbf{x}) \neq h'(\mathbf{x})] - \Pr_{\mathcal{D}_T}[h(\mathbf{x}) \neq h'(\mathbf{x})] \right|$, we have:*

$$\epsilon_{T_{t+1}}(h) \leq \frac{1}{\bar{\mu}} \left( \sum_{j=0}^{t} \mu^{t-j} \epsilon_{T_j}(h) + M \sum_{j=0}^{t} \mu^{t-j} \left( d_{\mathcal{H}\Delta\mathcal{H}}(\mathcal{D}_{T_j}, \mathcal{D}_{T_{t+1}}) + \frac{\lambda_j^*}{M} \right) \right)$$

*where $\lambda_j^* = \min_{h \in \mathcal{H}} \epsilon_{T_j}(h) + \epsilon_{T_{t+1}}(h)$.*

*(2) if discrepancy distance (Mansour et al., 2009) is adopted to measure the distribution shift, i.e., $d_{disc}(\mathcal{D}_{T_0}, \mathcal{D}_T) = \max_{h,h' \in \mathcal{H}} \left| \mathbb{E}_{\mathcal{D}_{T_0}}[\mathcal{L}(h(x), h'(x))] - \mathbb{E}_{\mathcal{D}_T}[\mathcal{L}(h(x), h'(x))] \right|$, we have:*

$$\epsilon_{T_{t+1}}(h) \leq \frac{1}{\bar{\mu}} \left( \sum_{j=0}^{t} \mu^{t-j} \epsilon_{T_j}(h) + \sum_{j=0}^{t} \mu^{t-j} \left( d_{disc}(\mathcal{D}_{T_j}, \mathcal{D}_{T_{t+1}}) + \Omega_j \right) \right)$$

*where $\Omega_j = \mathbb{E}_{\mathcal{D}_{T_j}}[\mathcal{L}(h_j^*(\mathbf{x}), y)] + \mathbb{E}_{\mathcal{D}_{T_{t+1}}}[\mathcal{L}(h_j^*(\mathbf{x}), h_{t+1}^*(\mathbf{x}))] + \mathbb{E}_{\mathcal{D}_{T_{t+1}}}[\mathcal{L}(h_{t+1}^*(\mathbf{x}), y)]$, and $h_j^* = \arg \min_{h \in \mathcal{H}} \epsilon_{T_j}(h)$ for $j = 0, \cdots, t, t+1$.*

The aforementioned domain discrepancy measures mainly focus on the marginal distribution over input features and have inspired a line of practical transfer learning algorithms (Ganin et al., 2016; Chen et al., 2019). However, recent work (Wu et al., 2019; Zhao et al., 2019) observed that the minimization of marginal distributions cannot guarantee the success of transfer learning in real scenarios. We propose to address this problem by incorporating the label information in the domain discrepancy measure (see next section).

## 4 LABEL-INFORMED DOMAIN DISCREPANCY

In this section, we introduce a novel label-informed domain discrepancy measure between the source domain $\mathcal{D}_{T_0}$ and target domain $\mathcal{D}_T$, its empirical estimate, and a transfer signature based on this measure to identify potential negative transfer. The use of this discrepancy measure in continuous transfer learning will be discussed in the next section.

### 4.1 $\mathcal{C}$-DIVERGENCE

For a hypothesis $h \in \mathcal{H}$, we denote $I(h)$ to be the subset of $\mathcal{X}$ such that $\mathbf{x} \in I(h) \Leftrightarrow h(\mathbf{x}) = 1$. In order to estimate the label-informed domain discrepancy from finite samples in practice, instead of Eq. (1), we propose the following $\mathcal{C}$-divergence between $\mathcal{D}_{T_0}$ and $\mathcal{D}_T$, taking into consideration the joint distribution over features and class labels:

$$d_{\mathcal{C}}(\mathcal{D}_{T_0}, \mathcal{D}_T) = \sup_{h \in \mathcal{H}} \left| \Pr_{\mathcal{D}_{T_0}}[\{I(h), y = 1\} \cup \{\overline{I(h)}, y = 0\}] - \Pr_{\mathcal{D}_T}[\{I(h), y = 1\} \cup \{\overline{I(h)}, y = 0\}] \right| \tag{2}$$

where $\overline{I(h)}$ is the complement of $I(h)$.

---

[2]In this case, we assume $\mu^0 = 1$ for any $\mu \geq 0$.

We show that some existing domain discrepancy methods (e.g., Ben-David et al. (2007)) can be seen as special cases of this definition by using the following relaxed covariate shift assumption.

**Definition 4.1.** (*Relaxed Covariate Shift Assumption*) The source and target domains satisfy the relaxed covariate shift assumption if for any $h \in \mathcal{H}$,
$$\Pr_{\mathcal{D}_{T_0}}[y \mid I(h)] = \Pr_{\mathcal{D}_T}[y \mid I(h)] = \Pr[y \mid I(h)] \tag{3}$$

Notice that it would be equivalent to covariance shift assumption (Shimodaira, 2000; Johansson et al., 2019) when $I(h)$ consists of only one example for all $h \in \mathcal{H}$ (see Lemma A.6 for details).

**Lemma 4.2.** *With the relaxed covariate shift assumption, for any $h \in \mathcal{H}$, we have:*
$$d_{\mathcal{C}}(\mathcal{D}_{T_0}, \mathcal{D}_T) = \sup_{h \in \mathcal{H}} \left| \left( Pr_{\mathcal{D}_{T_0}}[I(h)] - Pr_{\mathcal{D}_T}[I(h)] \right) \cdot \mathcal{S}_h + Pr_{\mathcal{D}_T}[y = 1] - Pr_{\mathcal{D}_{T_0}}[y = 1] \right|$$
*where $\mathcal{S}_h = Pr[y = 1|I(h)] - Pr[y = 0|I(h)]$.*

**Remark.** *From Lemma 4.2, we can see that in the special case where $\mathcal{S}_h$ is a constant for all $h \in \mathcal{H}$ and $Pr_{\mathcal{D}_T}[y = 1] = Pr_{\mathcal{D}_{T_0}}[y = 1]$, the proposed $\mathcal{C}$-divergence is reduced to the $\mathcal{A}$-distance (Ben-David et al., 2007) defined on the marginal distribution of features. More generally speaking, $\mathcal{C}$-divergence can be considered as a weighted version of the $\mathcal{A}$-distance where the hypothesis whose characteristic function has a larger class-separability (i.e., $|\mathcal{S}_h|$) receives a higher weight. Intuitively, compared to $\mathcal{A}$-distance, $\mathcal{C}$-divergence would pay less attention to class-inseparable regions in the input feature space, which provide irrelevant information for learning the prediction function in the target domain.*

Moreover, the following theorem states that in conventional transfer learning scenario with a static source domain and a static target domain, the target error is bounded in terms of $\mathcal{C}$-divergence across domains and the expected source error.

**Theorem 4.3.** *Assume that loss function $\mathcal{L}$ is bounded, i.e., there exists a constant $M > 0$ such that $0 \leq \mathcal{L} \leq M$. For a hypothesis $h \in \mathcal{H}$, we have the following bound:*
$$\epsilon_T(h) \leq \epsilon_{T_0}(h) + M \cdot d_{\mathcal{C}}(\mathcal{D}_{T_0}, \mathcal{D}_T)$$

### 4.2 EMPIRICAL ESTIMATE OF $\mathcal{C}$-DIVERGENCE

In practice, it is difficult to calculate the proposed $\mathcal{C}$-divergence based on Eq. (2) as it uses the true underlying distributions. Therefore, we propose the following empirical estimate of the $\mathcal{C}$-divergence between $\mathcal{D}_{T_0}$ and $\mathcal{D}_T$ as follows. Assuming that the hypothesis class $\mathcal{H}$ is symmetric (i.e., $1 - h \in \mathcal{H}$ if $h \in \mathcal{H}$), the empirical $\mathcal{C}$-divergence is:
$$d_{\mathcal{C}}(\hat{\mathcal{D}}_{T_0}, \hat{\mathcal{D}}_T) = 1 - \min_{h \in \mathcal{H}} \left| \frac{1}{m_{T_0}} \sum_{(\mathbf{x}, y): h(\mathbf{x}) \neq y} \mathbb{I}[(\mathbf{x}, y) \in \hat{\mathcal{D}}_0] + \frac{1}{m_T} \sum_{(\mathbf{x}, y): h(\mathbf{x}) = y} \mathbb{I}[(\mathbf{x}, y) \in \hat{\mathcal{D}}_T] \right| \tag{4}$$

where $\hat{\mathcal{D}}_{T_0}$ and $\hat{\mathcal{D}}_T$ denote the source and target domains with finite samples, respectively. $\mathbb{I}[a]$ is the binary indicator function which is 1 if $a$ is true, and 0 otherwise.

The following lemma provides the upper bound of the true $\mathcal{C}$-divergence using its empirical estimate.

**Lemma 4.4.** *For any $\delta \in (0, 1)$, with probability at least $1 - \delta$ over $m_{T_0}$ labeled source examples $\mathcal{B}_{T_0}$ and $m_T$ labeled target examples $\mathcal{B}_T$, we have:*

$$d_{\mathcal{C}}(\mathcal{D}_{T_0}, \mathcal{D}_T) \leq d_{\mathcal{C}}(\hat{\mathcal{D}}_{T_0}, \hat{\mathcal{D}}_T) + \left( \hat{\Re}_{\mathcal{B}_{T_0}}(L_H) + \hat{\Re}_{\mathcal{B}_T}(L_H) \right) + 3 \left( \sqrt{\frac{\log \frac{4}{\delta}}{2m_{T_0}}} + \sqrt{\frac{\log \frac{4}{\delta}}{2m_T}} \right)$$

*where $\hat{\Re}_{\mathcal{B}}(L_H)(\mathcal{B} \in \{\mathcal{B}_{T_0}, \mathcal{B}_T\})$ denotes the Rademacher complexity (Mansour et al., 2009) over $\mathcal{B}$ and $L_H = \{(\mathbf{x}, y) \to \mathbb{I}[h(\mathbf{x}) = y] : h \in \mathcal{H}\}$ is a class of functions mapping $\mathcal{Z} = \mathcal{X} \times \mathcal{Y}$ to $\{0, 1\}$.*

### 4.3 NEGATIVE TRANSFER CHARACTERIZATION

Informally, negative transfer is considered as the situation where transferring knowledge from the source domain has a negative impact on the target learner (Wang et al., 2019): $\epsilon_T(A(\mathcal{D}_{T_0}, \mathcal{D}_T)) > \epsilon_T(A(\emptyset, \mathcal{D}_T))$ where $A$ is the learning algorithm. $\epsilon_T$ is the target error induced by algorithm $A$. $\emptyset$ implies that it only considers the target data set for target learner. In this paper, we define a ***transfer signature*** to measure the transferability from source domain to target domain as follows.
$$TS(\mathcal{D}_T || \mathcal{D}_{T_0})) = \inf_{A \in \mathcal{G}} (\epsilon_T(A(\mathcal{D}_{T_0}, \mathcal{D}_T)) - \epsilon_T(A(\emptyset, \mathcal{D}_T))) \tag{5}$$
where $\mathcal{G}$ is the set of all learning algorithms. We state that source domain knowledge is not transferable over target domain when $TS(\mathcal{D}_T || \mathcal{D}_{T_0})) > 0$. Specially, since $A(\mathcal{D}_{T_0}, \mathcal{D}_T)$ learns an optimal classifier using both source and target data, we can define $\epsilon_T(A(\mathcal{D}_{T_0}, \mathcal{D}_T)) = \epsilon_T(h_\alpha^*)$

where $h_\alpha^* = \arg\min_{h \in \mathcal{H}(A)} \alpha \epsilon_T(h) + (1 - \alpha)\epsilon_{T_0}(h)$ and $\mathcal{H}(A)$ is the hypothesis space induced by $A$. When we only consider the target domain with $\alpha = 1$, $\epsilon_T(A(\emptyset, \mathcal{D}_T)) = \epsilon_T(h_T^*)$ where $h_T^* = \arg\min_{h \in \mathcal{H}(A)} \epsilon_T(h)$. Then we have the following theorem regarding the transfer signature.

**Theorem 4.5.** *Assuming the loss function $\mathcal{L}$ is bounded with $0 \leq \mathcal{L} \leq M$, we have*
$$\epsilon_T(h_\alpha^*) \leq \epsilon_T(h_T^*) + 2(1 - \alpha)M d_\mathcal{C}(\mathcal{D}_{T_0}, \mathcal{D}_T)$$

*Furthermore,*

$$TS(\mathcal{D}_T || \mathcal{D}_{T_0})) \leq 2(1 - \alpha)M d_\mathcal{C}(\mathcal{D}_{T_0}, \mathcal{D}_T)$$

**Remark.** *We have the following observations: (1) Larger $\mathcal{C}$-divergence between domains is often associated with a higher transfer signature, which indicates that negative transfer can be characterized using the proposed $\mathcal{C}$-divergence; (2) Empirically, the larger amount of labeled target data could increase the value of $\alpha$, resulting in the learned classifier relying more on the target data, which is consistent with the observation in (Wang et al., 2019). One extreme case is where $\alpha = 1$, implying we have adequate labeled target examples for standard supervised learning on the target domain without transferring knowledge from the source domain.*

## 5 PROPOSED ALGORITHM

In this section, we derive the continuous error bound based on our proposed $\mathcal{C}$-divergence, followed by a novel continuous transfer learning algorithm (CONTE) by minimizing the error upper bound. Notice that in the context of continuous transfer learning, we also use the proposed $\mathcal{C}$-divergence between the target domain at adjacent time stamps to measure the change in distribution over time.

### 5.1 CONTINUOUS ERROR BOUND WITH EMPIRICAL $\mathcal{C}$-DIVERGENCE

The following theorem states that for a bounded loss function $\mathcal{L}$, the target error in continuous transfer learning can be bounded in terms of the empirical classification error within source and historical target domains, the empirical $\mathcal{C}$-divergence across domains as well as the empirical Rademacher complexity of the class of functions $L_H = \{(\mathbf{x}, y) \to \mathbb{I}[h(\mathbf{x}) = y] : h \in \mathcal{H}\}$.

**Theorem 5.1.** *(Continuous Error Bound) Assume the loss function $\mathcal{L}$ is bounded with $0 \leq \mathcal{L} \leq M$. Given a source domain $\mathcal{D}_{T_0}$ and historical target domain $\{\mathcal{D}_{T_i}\}_{i=1}^t$, for $h \in \mathcal{H}$ and $\delta \in (0, 1)$, with probability at least $1 - \delta$, the target domain error $\epsilon_{T_{t+1}}$ on $\mathcal{D}_{T_{t+1}}$ is bounded as follows.*

$$\epsilon_{T_{t+1}}(h) \leq \frac{1}{\bar{\mu}} \left( \sum_{j=0}^t \mu^{t-j} \hat{\epsilon}_{T_j}(h) + M \sum_{j=0}^t \mu^{t-j} d_\mathcal{C}(\hat{\mathcal{D}}_{T_j}, \hat{\mathcal{D}}_{T_{t+1}}) + M\Lambda \right)$$

*where $\Lambda = \sum_{j=0}^t \left( \hat{\Re}_{\mathcal{B}_{T_j}}(L_H) + \hat{\Re}_{\mathcal{B}_{T_{t+1}}}(L_H) + 3\sqrt{\frac{\log \frac{8}{\delta}}{2m_{T_j}}} + 3\sqrt{\frac{\log \frac{8}{\delta}}{2m_{T_{t+1}}}} + \sqrt{\frac{M^2 \log \frac{4}{\delta}}{2m_{T_j}}} \right)$.*

**Remark.** *Compared to continuous error bounds in Corollary 3.2 using existing domain divergence measures (Ben-David et al. (2007); Mansour et al. (2009)), our bound consists of only data-dependent terms (e.g., empirical source error and $\mathcal{C}$-divergence), whereas previous error bounds are determined by the error terms involving the intractable labeling function or optimal target hypothesis (see Corollary 3.2).*

### 5.2 CONTE ALGORITHM

For continuous transfer learning, we leverage both the source domain and historical target domain data to learn the predictive function for the current time stamp. To this end, we propose to minimize the error bound in Theorem 5.1 for learning the predictive function on $\mathcal{D}_{T_{t+1}}$. Furthermore, we aim to learn a domain-invariant and time-invariant latent feature space such that the $\mathcal{C}$-divergence across domains and across time stamps could be minimized. Therefore, we present an adversarial Variational Auto-encoder (VAE) algorithm with the following overall objective function:

$$\mathcal{J}(T_0, T_1, T_2, \cdots, T_{t+1}) = \sum_{j=0}^t \mu^{t-j} \left( \mathcal{L}_{clc}(T_j, T_{t+1}) + d_\mathcal{C}(\hat{\mathcal{D}}_{T_j}, \hat{\mathcal{D}}_{T_{t+1}}) + \lambda \mathcal{L}_{ELBO}(T_j, T_{t+1}) \right) \quad (6)$$

where $\mathcal{L}_{clc}(T_j, T_{t+1})$ represents the classification error over the labeled examples from $\mathcal{D}_{T_j}$ and $\mathcal{D}_{T_{t+1}}$, $d_\mathcal{C}(\hat{\mathcal{D}}_{T_j}, \hat{\mathcal{D}}_{T_{t+1}})$ is the empirical estimate of $\mathcal{C}$-divergence across domain. Thus the first two terms of Eq. (6) are associated with $\hat{\epsilon}_{T_j}(h) + d_\mathcal{C}(\hat{\mathcal{D}}_{T_j}, \hat{\mathcal{D}}_{T_{t+1}})$ in the error bound of Theorem 5.1. The third term $\mathcal{L}_{ELBO}(T_j, T_{t+1})$ is the variational bound in the VAE framework (see Figure 4) when learning the latent feature space and $\lambda > 0$ is a hyper-parameter. In this case, we have $\mu \in [0, 1]$ because we assume that the data distribution of a time-evolving target domain shifts smoothly over time. Then we instantiate the terms of Eq. (6) as follows.

Inspired by semi-supervised VAE (Kingma et al., 2014), we propose to learn the feature space by maximizing the following likelihood across domains.

$$\log p_\theta(\mathbf{x}, y) = \mathrm{KL}\big(q_\phi(\mathbf{z}|\mathbf{x}, y)||p_\theta(\mathbf{z}|\mathbf{x}, y)\big) + \mathbb{E}_{q_\phi(\mathbf{z}|\mathbf{x}, y)}[\log p_\theta(\mathbf{x}, y, \mathbf{z}) - \log q_\phi(\mathbf{z}|\mathbf{x}, y)]$$

where $\phi$ and $\theta$ are the learnable parameters in the encoder and decoder respectively, and $\mathbf{z}$ is the latent feature representation of the input example $(\mathbf{x}, y)$. $\mathrm{KL}(\cdot||\cdot)$ is Kullback–Leibler divergence. The evidence lower bound (ELBO), a lower bound on this log-likelihood, can be written as follows.

$$\mathcal{E}_{\theta,\phi}(\mathbf{x}, y) = \mathbb{E}_{q_\phi(\mathbf{z}|\mathbf{x}, y)}\left[\log p_\theta(\mathbf{x}, y|\mathbf{z})\right] + \mathrm{KL}\left(q_\phi(\mathbf{z}|\mathbf{x}, y)||p(\mathbf{z})\right) \tag{7}$$

where $\mathcal{E}_{\theta,\phi}(\mathbf{x}, y) \leq \log p_\theta(\mathbf{x}, y)$. Similarly, we have the following ELBO to maximize the log-likelihood of $p_\theta(\mathbf{x})$ when the label is not available:

$$\mathcal{U}_{\theta,\phi}(\mathbf{x}) = \sum_y \left(q_\phi(y|\mathbf{x}) \cdot \mathcal{E}_{\theta,\phi}(\mathbf{x}, y) - \mathbb{E}_{q_\phi(y|\mathbf{x})}\left[\log q_\phi(y|\mathbf{x})\right]\right) \tag{8}$$

where $p_\theta(\mathbf{x}, y, \mathbf{z}) = p_\theta(\mathbf{x}|y, \mathbf{z})p_\theta(y|\mathbf{z})p(\mathbf{z})$ with prior Gaussian distribution $p(\mathbf{z}) = \mathcal{N}(\mathbf{0}, \mathbf{I})$. Therefore, the ***variational bound*** $\mathcal{L}_{ELBO}(T_j, T_{t+1})$ is given below.

$$\mathcal{L}_{ELBO}(T_j, T_{t+1}) = -\sum_{i=1}^{m_{T_j}+m_{T_{t+1}}} \mathcal{E}_{\theta,\phi}(\mathbf{x}_i, y_i) - \sum_{i=1}^{u_{T_{t+1}}} \mathcal{U}_{\theta,\phi}(\mathbf{x}_i, y_i) \tag{9}$$

where $u_{T_{t+1}}$ is the number of unlabeled training examples from $\mathcal{D}_{T_{t+1}}$. Besides, the ***classification error*** $\mathcal{L}_{clc}(T_j, T_{t+1})$ can be expressed as follows.

$$\mathcal{L}_{clc}(T_j, T_{t+1}) = \sum_{i=1}^{m_{T_j}+m_{T_{t+1}}} \mathcal{L}\left(y_i, q_\phi(\cdot|\mathbf{x}_i)\right) \tag{10}$$

where $q_\phi(\cdot)$ is the discriminative classifier formed by the distribution $q_\phi(y|\mathbf{x})$ in Eq. (8), and $\mathcal{L}(\cdot, \cdot)$ is the cross-entropy loss function in our experiments. To estimate the $\mathcal{C}$-divergence, we first define $\tilde{h}$ to be a two-dimensional characteristic function with $\tilde{h}(\mathbf{x}, y) = 1 \Leftrightarrow h(\mathbf{x}) = y \Leftrightarrow \{h(\mathbf{x}) = 1, y = 1\} \vee \{h(\mathbf{x}) = 0, y = 0\}$ for $h \in \mathcal{H}$. Then the empirical $\mathcal{C}$-divergence in Eq. (4) can be rewritten as follows.

$$d_\mathcal{C}(\hat{\mathcal{D}}_{T_j}, \hat{\mathcal{D}}_{T_{t+1}}) = 1 - \min_{\tilde{h}} \left| \frac{1}{m_{T_j}} \sum_{(\mathbf{x},y):\tilde{h}(\mathbf{x},y)=0} \mathbb{I}[(\mathbf{x}, y) \in \hat{\mathcal{D}}_{T_j}] + \frac{1}{m_{T_{t+1}}} \sum_{(\mathbf{x},y):\tilde{h}(\mathbf{x},y)=1} \mathbb{I}[(\mathbf{x}, y) \in \hat{\mathcal{D}}_{T_{t+1}}] \right|$$

Note that the latent feature representation $\mathbf{z}$ learned by $q_\phi(\mathbf{z}|\mathbf{x}, y)$ could capture the label-informed information of an example $(\mathbf{x}, y)$. Thus, the hypothesis $\tilde{h}$ can be considered as the composition of a feature extraction $q_\phi$ and a domain classifier $\mathcal{F}_j$, i.e. $\tilde{h}(\mathbf{x}, y) = \mathcal{F}_j(q_\phi(\mathbf{z}|\mathbf{x}, y))$. Formally, the ***empirical estimate of*** $\mathcal{C}$***-divergence*** is given below.

$$d_\mathcal{C}(\hat{\mathcal{D}}_{T_j}, \hat{\mathcal{D}}_{T_{t+1}}) = 1 - \min_{\mathcal{F}_j} \left| \frac{1}{m_{T_j}} \sum_{\mathbf{z}:\mathcal{F}_j(\mathbf{z})=0} \mathbb{I}[\mathbf{z} \in \hat{\mathcal{D}}_{T_j}] + \frac{1}{m_{T_{t+1}}} \sum_{\mathbf{z}:\mathcal{F}_j(\mathbf{z})=1} \mathbb{I}[\mathbf{z} \in \hat{\mathcal{D}}_{T_{t+1}}] \right| \tag{11}$$

The benefits of CONTE are twofold: first, it learns the latent feature space using both input $\mathbf{x}$ and output $y$; second, it minimizes a tighter error upper bound based on $\mathcal{C}$-divergence in Theorem 5.1. This framework can also be interpreted as a minimax game: the VAE learns a domain-invariant and time-invariant latent feature space, whereas the domain classifier $\mathcal{F}_j$ aims to distinguish the examples from different domains and different time stamps. In this paper, we adopt the gradient reversal layer (Ganin et al., 2016) when updating the parameters of domain classifier $\mathcal{F}_j$, and thus CONTE can be optimized by back-propagation in an end-to-end manner (see Algorithm 1 in appendices).

However, we observe that (1) it is difficult to estimate the $\mathcal{C}$-divergence with only limited labeled target examples from $\mathcal{D}_{T_{t+1}}$; (2) when learning the latent features $\mathbf{z}$, combining the data $\mathbf{x}$ (e.g., one image) and class-label $y$ directly might lead to over-emphasizing the data itself due to its high dimensionality compared to $y$. To address these problems, we propose the following *Pseudo-label Inference*, i.e., we infer the pseudo labels of unlabeled examples using the classifier $q_\phi(y|\mathbf{x})$ for each training epoch. Using labeled source and target examples as well as unlabeled target examples with inferred pseudo labels, the $\mathcal{C}$-divergence could be estimated in a balanced setting. Furthermore, to enforce the compatibility between features $\mathbf{x}$ and label $y$, we adopt a pre-encoder step to learn a dense representation for the input $\mathbf{x}$, and then learn the label-informed latent features $\mathbf{z}$.

# 6 EXPERIMENTAL RESULTS

**Synthetic Data:** We generate a synthetic data set in which each domain has 1000 positive examples and 1000 negative examples randomly generated from Gaussian distributions $\mathcal{N}([1.5\cos\theta, 1.5\sin\theta]^T, 0.5 \cdot \mathbf{I}_{2\times2})$ and $\mathcal{N}([1.5\cos(-\theta), 1.5\sin(-\theta)]^T, 0.5 \cdot \mathbf{I}_{2\times2})$, respectively. We let $\theta = 0$ for the source domain (denoted as S1), and $\theta = \frac{i \cdot \pi}{t}(i = 1, \cdots, t)$ for the time evolving target domain with $t = 8$ time stamps (denoted as T1, $\cdots$, T8).

**Image Data:** We consider the following two tasks: digital classification (MNIST, SVHN) and image classification (Office-31 with three domains: Amazon, DSLR and Webcam; and Office-Home with

four domains: Art, Product, Clipart and Real World). Since standard domains are static in these data sets, we will simulate the time-evolving distribution shift on the target domain by adding noise (e.g., random salt&pepper noise, adversarial noise, rotation). Take SVHN→MNIST as an example, we will use SVHN as the static source domain, and MNIST as the target domain at the first time stamp. By adding adversarial noise to the MNIST images, we obtain a time-evolving target domain (denoted as T1, $\cdots$, T11 in Table 1). For Office-31 and Office-Home, we add the random salt&pepper noise and rotation to generate the evolving target domain. More details can be found in the appendices.

**Baselines:** The baseline methods are as follows. (1) SourceOnly: training with only source data; (2) TargetERM: empirical risk minimization (ERM) on only target domain; (3) DAN (Long et al., 2015), CORAL (Sun & Saenko, 2016), DANN (Ganin et al., 2016), ADDA (Tzeng et al., 2017), WDGRL (Shen et al., 2018), DIFA (Volpi et al., 2018) and MDD (Zhang et al., 2019): training with feature distribution alignment. (4) CONTE: training with label-informed distribution alignment on the evolving target domain while $\mu \in \{0, 0.2, 0.4, 0.6, 0.8, 1\}$. (5) CONTE$_\infty$: a one-time transfer learning variant of CONTE that directly transfers from source domain to current target domain. We fix $\lambda = 0.1$, and all the methods use the same neural network architecture for feature extraction.

### 6.1 EVALUATION OF $\mathcal{C}$-DIVERGENCE

We compare the proposed $\mathcal{C}$-divergence with conventional domain discrepancy measure $\mathcal{A}$-distance (Ben-David et al., 2007) on a synthetic data set with an evolving target domain. We assume that the hypothesis space $\mathcal{H}$ consists of linear classifiers in the feature space. Figure 2 shows the domain discrepancy and target classification accuracy for each pair of source and target domains. We have the following obser-

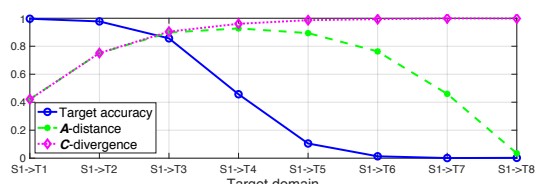

Figure 2: Comparison of domain discrepancy and target accuracy

vations. (1) The classification accuracy on the target domain significantly decreases from target domain T1 to T8. One explanation is that the joint distribution $p(x, y)$ on the time evolving target domain gradually shifted. (2) The $\mathcal{A}$-distance increases from S1→T1 to S1→T4, and then decreases from S1→T4 to S1→T8. That is because it only estimates the difference of the marginal feature distribution $p(x)$ between the source and target domains. (3) The $\mathcal{C}$-divergence keeps increasing from S1→T1 to S1→T8, which indicates the decreasing task relatedness between the source and the target domains. Therefore, compared with $\mathcal{A}$-distance[3], the proposed $\mathcal{C}$-divergence better characterizes the transferability from the source to the target domains.

### 6.2 EVALUATION OF ERROR BOUND

When there is only one time stamp involved in the target domain, Theorem 5.1 is reduced to the standard error bound in the conventional static transfer learning setting. We empirically compare this reduced error bound with the existing Rademacher complexity based error bound in (Mansour et al., 2009) (see Theorem A.4 in appendices for being self-contained).

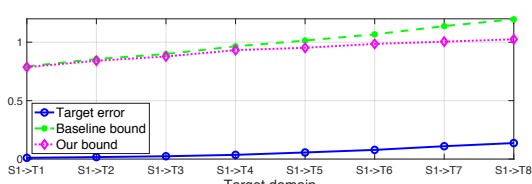

Figure 3: Comparison of error bounds

We use the 0-1 loss function as $\mathcal{L}$ and assume that the hypothesis space $\mathcal{H}$ consists of linear classifiers in the feature space. Figure 3 shows the estimated error bounds and target error with the time evolving target domain (i.e., S1→T1, $\cdots$, S1→T8 in a new synthetic data set with a slower time evolving target domain to ensure that the baseline bound is meaningful most of the time) where we choose $h = h_{T_0}^*$. It demonstrates that our $\mathcal{C}$-divergence based error bound is much tighter than the baseline. Notice that when transferring source domain S1 to target domain T8, our error bound is largely determined by the $\mathcal{C}$-divergence, whereas the baseline is determined by the difference between the optimal source and target hypothesizes. Furthermore, given any hypothesis $h \in \mathcal{H}$, we may not be able to estimate the baseline bound when the optimal hypothesis is not available.

### 6.3 EVALUATION OF CONTINUOUS TRANSFER LEARNING

Tables 1 and 2 provide the continuous transfer learning results on digital and office-31 data sets where the classification accuracy on target domain is reported (the best results are highlighted in bold). It is observed that (1) the classification accuracy using SourceOnly algorithm significantly

---

[3]The results for other existing discrepancy measures follow a similar pattern and thus omitted for brevity

decreases on the evolving target domain due to the shift of joint data distribution $p(\mathbf{x}, y)$ on target domain; (2) the performance of static baseline algorithms is largely affected by the distribution shift in the evolving target domain, and even worse than TargetERM in some cases (e.g., on T6-T11 from SVHN to evolving MNIST); (3) CONTE significantly outperforms CONTE$_\infty$ as well as other competitors on target domain by a large margin (i.e., up to 30% improvement on the last time stamp of target domain) because it effectively leverages the historical target domain information to smoothly re-align the target distribution when the change of target domain distribution in consecutive time stamps is small.

Table 1: Transfer learning accuracy from SVHN (source) to time evolving MNIST (target)

| Target Domain | T1 | T2 | T3 | T4 | T5 | T6 | T7 | T8 | T9 | T10 | T11 |
|---|---|---|---|---|---|---|---|---|---|---|---|
| SourceOnly | 0.6998 | 0.6738 | 0.6336 | 0.5692 | 0.4747 | 0.4110 | 0.3087 | 0.2220 | 0.1481 | 0.0828 | 0.0764 |
| TargetERM | 0.7451 | 0.6997 | 0.6618 | 0.6314 | 0.6368 | 0.6359 | 0.6695 | 0.7133 | 0.7214 | 0.7450 | 0.7512 |
| CORAL | 0.8349 | 0.8410 | 0.7633 | 0.7063 | 0.6496 | 0.5900 | 0.5031 | 0.5101 | 0.4337 | 0.4156 | 0.4502 |
| DANN | 0.8666 | 0.8356 | 0.8018 | 0.7529 | 0.7309 | 0.6641 | 0.6614 | 0.5618 | 0.5204 | 0.5082 | 0.4594 |
| ADDA | 0.8667 | 0.8487 | 0.7982 | 0.7187 | 0.6804 | 0.5397 | 0.4366 | 0.3473 | 0.2636 | 0.1659 | 0.1259 |
| WDGRL | 0.8990 | 0.8602 | 0.8247 | 0.8222 | 0.7452 | 0.6877 | 0.6481 | 0.5896 | 0.5145 | 0.4952 | 0.5196 |
| DIFA | 0.9164 | 0.8993 | 0.8713 | 0.8273 | 0.7935 | 0.6661 | 0.5956 | 0.4381 | 0.3479 | 0.2448 | 0.1332 |
| CONTE$_\infty$ | **0.9747** | 0.9552 | 0.9514 | 0.9279 | 0.8801 | 0.8833 | 0.8691 | 0.6979 | 0.7030 | 0.7415 | 0.7316 |
| CONTE | **0.9747** | **0.9740** | **0.9803** | **0.9864** | **0.9908** | **0.9940** | **0.9950** | **0.9965** | **0.9970** | **0.9967** | **0.9975** |

Table 2: Transfer learning accuracy on Office-31

| | Amazon → Webcam | | | | | Webcam → DSLR | | | | |
|---|---|---|---|---|---|---|---|---|---|---|
| | T1 | T2 | T3 | T4 | T5 | T1 | T2 | T3 | T4 | T5 |
| SourceOnly | 0.7490 | 0.2255 | 0.2282 | 0.1275 | 0.1503 | 0.9651 | 0.4309 | 0.3329 | 0.1611 | 0.2027 |
| TargetERM | 0.5584 | 0.3933 | 0.4215 | 0.3396 | 0.3732 | 0.4966 | 0.4201 | 0.4188 | 0.3248 | 0.4067 |
| DAN | 0.8537 | 0.5007 | 0.4993 | 0.3638 | 0.4470 | 0.9772 | 0.7302 | 0.6161 | 0.4765 | 0.5302 |
| CORAL | 0.8711 | 0.5235 | 0.4819 | 0.3195 | 0.4054 | 0.9812 | 0.7289 | 0.6671 | 0.4846 | 0.5221 |
| DANN | 0.8389 | 0.4993 | 0.4121 | 0.3973 | 0.3382 | 0.9651 | 0.7356 | 0.6416 | 0.4510 | 0.5490 |
| MDD | 0.8940 | 0.6738 | 0.5490 | 0.5141 | 0.4295 | 0.9724 | 0.8738 | 0.7315 | 0.5047 | 0.5289 |
| CONTE$_\infty$ | **0.9154** | 0.6376 | 0.5758 | 0.4591 | 0.4846 | 0.9785 | 0.8591 | 0.7289 | 0.4926 | 0.5557 |
| CONTE | **0.9154** | **0.8134** | **0.8081** | **0.7611** | **0.7826** | 0.9785 | **0.9235** | **0.9208** | **0.8886** | **0.9154** |

## 7 RELATED WORK

**Transfer Learning:** Transfer learning (Ying et al., 2018; Jang et al., 2019) improves the performance of a learning algorithm on the target domain by using the knowledge from the source domain. It is theoretically proven that the target error is well bounded (Ben-David et al., 2010; Mansour et al., 2009), followed by a line of practical algorithms (Shen et al., 2018; Long et al., 2017; 2018; Saito et al., 2018; Chen et al., 2019) with covariate shift assumption. However, it is observed that this assumption does not always hold in real-world scenarios (Rosenstein et al., 2005; Wang et al., 2019).

**Multi-source Domain Adaptation:** Multi-source domain adaptation improves the target prediction function from multiple source domains (Zhao et al., 2018; Hoffman et al., 2018; Wen et al., 2020). It is similar to our problem setting as source and historical target domains can be considered as multiple "source" domains when modeling the target domain at the current time stamp. However, only limited labeled target examples are provided in our problem setting, whereas multi-source domain adaptation requires that all source domains have adequate labeled examples.

**Continual Learning:** Continual lifelong learning (Parisi et al., 2019; Rusu et al., 2016; Hoffman et al., 2014; Bobu et al., 2018) involves the sequential learning tasks with the goal of learning a predictive function on the new task using knowledge from historical tasks. Most of them focused on mitigating catastrophic forgetting when learning new tasks from only one evolving domain, whereas our work studied the transferability between a source domain and a time evolving target domain.

## 8 CONCLUSION

In this paper, we study continuous transfer learning with a time evolving target domain, which has not been widely studied and yet is commonly seen in many real applications. We start by deriving a generic error bound of continuous transfer learning with flexible domain discrepancy measures. Then we propose a novel label-informed $\mathcal{C}$-divergence to measure the domain discrepancy incorporating the label information, and study its application in continuous transfer learning, which leads to an improved error bound. Based on this bound, we further propose a generic adversarial Variational Auto-encoder algorithm named CONTE for continuous transfer learning. Extensive experiments on both synthetic and real data sets demonstrate the effectiveness of our CONTE algorithm.

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
