# OpenReview forum: "Continuous Transfer Learning"
_ICLR.cc/2021/Conference — Reject_

### Official Review · AnonReviewer3 · 2020-10-29
**This paper focuses on the continuous transfer learning setting with a time evolving target domain. A major challenge is that the relationship between the source domain and the target domain changes over time. This paper proposes a new C-divergence to measure the discrepancy between domains. It could be utilized to instantiate a tighter error upper bound in the continuous transfer learning setting. The clarity of this paper still needs some improvement. And experimental results are insufficient.**

**Rating:** 6
**Confidence:** 4

**Review:**


(1) This article is more detailed in its theoretical analysis, and the C-divergence has advantages over other divergence from the theoretical analysis.

(2) This article proposes a new continuous transfer learning setting. It theoretically analyzes the upper bound of the target domain error, and utilizes the C-divergence to measure the joint distribution discrepancy in the two domains.

(3) Some parts of the presentation needs to be improved, such as the presentation of negative transfer in section 4.3. How to use other methods such as DAN to this new setting? It should be described in more details in section 6. How to set the target domain label in section 6?

(4) Although the results outperforms some other methods, the experiments are not sufficient. In experiments, the authors only do experiments in some easy task in DA, and no experiments are shown for the effect of the hyper-parameter mu.
Detailed Evaluation:

(5) About formula (10), why does the cross-entropy of T_{t+1} calculate t+1 times?

---

> ### Author Response · Authors · 2020-11-17
> **Response to AnonReviewer3**
>
> Thank you very much for the comments.
>
> (1) Q: Some parts of the presentation need to be improved, such as the presentation of negative transfer in section 4.3. How to use other methods such as DAN to this new setting? It should be described in more details in section 6. How to set the target domain label in section 6?
>
> A: Due to the limited pages of the submission, we described the detailed experimental setting in our Appendix A.4. More specifically, other methods such as DAN typically learn the target predictive function over T_{t+1} using knowledge only from the source task. But in our experiments, we let them use both adequate source labeled examples and limited target labeled examples in the learning process. In Figure 8 in the appendix, we show how the limited label information in the target domain helps alleviate negative transfer for those baseline methods. The target domain label setting can be found in Appendix A.4.1 for different data sets.
>
> (2) Q: Although the results outperform some other methods, the experiments are not sufficient. In experiments, the authors only do experiments in some easy task in DA, and no experiments are shown for the effect of the hyper-parameter mu.
>
> A: We would like to clarify that we did conduct additional experiments, including performance evaluation on other image data sets and hyperparameter analysis (e.g., \mu), in our Appendix A.4.3 (could be found in the supplemental material). More specifically, we empirically analyzed the selection of hyper-parameter \mu associated with the C-divergence.
>
> (3) Q: About formula (10), why does the cross-entropy of T_{t+1} calculate t+1 times?
>
> A: We would like to point out that actually the cross-entropy of T_{t+1} only needs to be calculated once. Combing the Eq. (10) and Eq. (6), the overall cross-entropy of T_{t+1} is \sum_{j=0}^t \mu^{t-j}L_{clc}(T_{t+1}) = L_{clc}(T_{t+1}) because \sum_{j=0}^t \mu^{t-j} = 1 with normalized \mu on every historical task.

---

### Official Review · AnonReviewer4 · 2020-11-01
**Interesting question and results, but easily generalizable**

**Rating:** 5
**Confidence:** 4

**Review:**

Summary:

This paper studies how to transfer the information in the static source domain to the time-evolving target domain. This paper proposes a domain discrepancy measure and an algorithm for continuous transfer learning. The results seem to be interesting and the problem this paper studies is important. However, the domain rate in the main results and algorithm could be easily generalized which can make the results more broadly applicable. Moreover, it needs more clarification about the motivation of using the C-divergence measure in the time-evolving target domain.

Major comments:

1. Domain decay rate: There is a domain decay rate in all the main results and the algorithm. From the proof of all the main results, the rate $\mu^{t-j}/\bar{\mu}$ can be easily generalized to any arbitrary weight $\omega_j$ with the constraint $\sum_{j = 0}^t \omega_j = 1$ and $\omega_j \geq 0$. Hence, there is no assumption about $\mu$ in the loss function. Allowing general $\omega_j$ is more appealing for many reasons.

First, this paper indexes the source domain by $T_0$. The CONTE algorithm assigns similar weights to the source domain ($\mu^t$) and target domain at time 1 ($\mu^{t-1}$), but a different weight to the target domain at time $t$ ($\mu^0$) (when $\mu \neq 1$). This algorithm seems to implicitly assume the target domain at time 1 is more similar to the source domain than the target domain at time $t$, which seems unnatural and requires further justification. However, using  $\omega_j$ does not have this issue.

Second, it is possible that the time-evolving domain has a cyclic or seasonality pattern. Even in the experiment setup in Section 6, $\sin(\theta) = \sin(\frac{i\cdot \pi}{t})$ is cyclic. It may make more sense to assign a larger weight to the time period whose domain is more similar to the current domain.

2. Potential improvement of the algorithm: Following the previous point, the CONTE algorithm could potentially be improved. Since the domain discrepancy measure $d_c(\cdot, \cdot)$ can be estimated, we could assign the weight $\omega_j$ inverse proportional to $d_c(\cdot, \cdot)$ and the error could be potentially reduced.

3. C-divergence measure: It seems the C-divergence measure does not involve the time concept and is generally applicable to transfer learning tasks beyond continuous transfer learning? Then why is the C-divergence measure particularly of interest in the context of continuous transfer learning and why is it more preferred to use the C-divergence measure rather than the conventional measures, such as A-divergence in this setting? Some discussions on this point and numerical comparisons will be helpful.

---

> ### Author Response · Authors · 2020-11-17
> **Response to AnonReviewer4**
>
> Thanks very much for your comments
>
> (1) Q: Allowing general ωj is more appealing? Since the domain discrepancy measure dc(⋅,⋅) can be estimated, we could assign the weight ωj inverse proportional to dc(⋅,⋅) and the error could be potentially reduced?
>
> A: We adopted the domain weight decay rate for the following reasons. (1) In this paper, we assume that the target domain is continuously evolving over time. It is a natural idea to learn more knowledge from the recent task, especially when the relatedness of the current task and older tasks is unknown. (2) Indeed, the time-evolving domain might have a cyclic or seasonality pattern. It still makes sense to use more knowledge from the recent task due to the continuous evolving assumption. Besides, intuitively, automatically learned weight \mu might improve the model performance. But the additional challenges are that (i) it largely depends on the weight estimated method; (ii) it would induce additional computational time complexity, which may be prohibitive for the dynamic setting. In fact, we have attempted to incorporate some existing weight estimation methods from multi-source domain adaptation algorithms (e.g., MDAN [1] and DARN [2]) into our CONTE algorithm. The results are given below on Office-31 (Webcam->DSLR). It is shown that the automatically learned weight values of \mu might lead to inferior predictive performance compared to the domain weight decay rate strategy. Recent research work [3] on multi-source domain adaptation has shown that simply assigning the weight ωj inverse proportional to domain discrepancy might not be optimal. This indicates that exactly estimating the weight of each historical task over the current task is much more challenging, and the sub-optimal solution is likely to lead to inferior performance. In contrast, the strategy with the domain weight decay rate enjoys better performance and high model efficiency.
>
>
>                             | T1 | T2 | T3 | T4 | T5
> CONTE               | 0.9785 | 0.9235 | 0.9208 | 0.8886 | 0.9154
>
> CONTE_MDAN  |  0.9785 | 0.9356 | 0.8859 | 0.8027 | 0.7964
>
> CONTE_DARN   | 0.9785 | 0.9208 | 0.8644 | 0.7664 | 0.7463
>
> [1] Zhao, Han, Shanghang Zhang, Guanhang Wu, José MF Moura, Joao P. Costeira, and Geoffrey J. Gordon. "Adversarial multiple source domain adaptation." In Advances in neural information processing systems, pp. 8559-8570. 2018.
>
> [2] Wen, Junfeng, Russell Greiner, and Dale Schuurmans. "Domain Aggregation Networks for Multi-Source Domain Adaptation." In ICML, 2020.
>
> [3] Mansour, Yishay, Mehryar Mohri, Ananda Theertha Suresh, and Ke Wu. "A Theory of Multiple-Source Adaptation with Limited Target Labeled Data." arXiv preprint arXiv:2007.09762(2020).
>
> (2) Q: why is the C-divergence measure particularly of interest in the context of continuous transfer learning and why is it more preferred to use the C-divergence measure rather than the conventional measures, such as A-divergence in this setting?
>
> A: We would like to point out that C-divergence is more preferred in our problem setting for the following reasons. (i) Theoretically, as shown in Theorem 5.1, C-divergence could be used to derive much tighter continuous transfer learning error bound, which has also been empirically confirmed in Section 6.1 compared to conventional A-divergence. (ii) It is indeed applicable to the general transfer learning setting. But compared to A-divergence (over only marginal feature distribution), C-divergence is more suitable for measuring the domain discrepancy in the semi-supervised setting by leveraging both feature and label information across tasks. We empirically compared the C-divergence with A-divergence in Figure 10 in the appendix (shown in our supplemental material), which confirms the better performance of C-divergence over A-divergence in the semi-supervised setting.

---

### Official Review · AnonReviewer2 · 2020-11-02
**Interesting problem on Non-stationary Transfer Learning**

**Rating:** 6
**Confidence:** 4

**Review:**


The paper proposed a transfer learning setting where the target domain varies/evolves over time and the source domain is considered static. The paper uses C-divergence to measure label-dependent domain discrepancy between source/previous target domain and the current target domain and provided a theoretical bound. The paper also used supervised VAE for CONTE algorithm and included C-divergence as a part of the objective function.

The paper address an interesting and an important practical problem on non-stationary target distribution. The proposed (semi-supervised) divergence measure and it’s relation to the previously proposed (unsupervised) divergence measures such as (A-distance and disc distance) is well-motivated. Please consider the following questions to help me understand the problem.

### Major Questions:

* The reason for the label dependency is motivated as explained but I believe that the usage of adversarial semi-supervised VAE was not properly motivated and related it the C-divergence.

* Sec 5.2 needs a little clarification for us readers to understand it clearly.  For instance, How is minimizing Eq 6 minimizes Continuous Error bound in Theorem 5.1. The first two terms in Eq 6 is related to the bound but Why is the ELBO term needed and how it is related to the bound. Eq 7-9 follows directly from the Kingma et al.’s Semi-supervised VAE so no explanation is needed.

* Please help me to understand how to get Eq 4 from Lemme 4.2. I couldn’t find the related steps in the Suppl. On a related question, it seems that, in Eq 11 or the one above … if the F-classifier doesn’t distinguish between the domains (across the time or between source and target domains), ie., if F-classifier always say  that the example is from one of the two domains, then one of the two terms in the bracket will be 0 and the other will be 1 and it will make the distance measure to 0.

* I believe the problem setting is very closely related to the Continual learning setting,  addition. To the Catastrophic forgetting, most recent work in Continual learning focuses on Negative transfer (such as Gradient Episodic Memory). The paper will be stronger if the key comparison to baselines in Continual learning is added.

### Minor Questions:

* Notation issue in Eq.9 for the unlabeled U(x) to estimate the log-likelihood of the marginal distribution of the features. ‘y’ shouldn’t be in the U function.

* Overloaded use of notation F in Eq. 11 and in Suppl.

* In many practical applications, for instance product reviews, if a new version of the product released periodically with new sets of features. Even though, the review data is  not only from the same target distribution but sample the different regions of this same target distribution. Can the current distance measure and  the CONTE algorithm be adapted to this problem setting.

* Related question to the one above: Can we automatically learn \mu from the data? If there is transfer from Ti to Tt+1 but not from Tt to Tt+1, will the proposed method can be adapted to this problem setting. Does the information from Ti is retained in Tt from previous learning cycle, such that, it will be used by Tt+1 task?

i have read the author(s)' comments and I have updated the ratings based on their replies. Thanks for very extensive clarification. Adding these comments in the final revision would significantly improve the quality of the paper.

---

> ### Author Response · Authors · 2020-11-17
> **Response to AnonReviewer2**
>
> Thank you very much for the comments.
>
> (1) Q: how semi-supervised VAE relate to C-divergence?
>
> A: Thank you very much for the comments. The semi-supervised VAE would learn a label-informed feature representation for each example such that our C-divergence could then be empirically estimated from the label-informed features across domains. The benefit of this labeled-informed feature representation is to reduce the empirical C-divergence across domains, compared to the estimate of C-divergence on the original input space directly. It has been observed in previous work [1,2] that learning a new low-dimensional feature common space could reduce the domain divergence across domains. The key difference between our work and previous work [1,2] is that previous work aims to learn the marginal feature space mapping from only the input space, whereas we need the semi-supervised VAE to learn the label-informed feature space mapping from both the input space and class label space.
>
> [1] Wang, Zheng, et al. "Transferred dimensionality reduction." In Joint European conference on machine learning and knowledge discovery in databases, pp. 550-565. 2008.
> [2] Ganin, Yaroslav, et, al "Domain-adversarial training of neural networks." The Journal of Machine Learning Research, 2016.
>
> (2) Q: relation of Eq 6 to Theorem 5.1?
>
> A: It is clear that the first two terms in Eq 6 are directly adopted from the bound, and the ELBO term is added to regularize the latent feature space within which we estimate the empirical C-divergence. To be more specific, the ELBO term aims to learn the label-informed feature representation because C-divergence is defined to measure the shift of joint distribution over both features and label spaces. Then the C-divergence is empirically estimated in the label-informed feature space regularized by the ELBO term, which is fundamentally different from previous domain discrepancy measures (e.g., A-distance), which is estimated from the marginal feature space without label information.
>
> (3) Q: relation to Kingma et al.’s paper?
>
> A: Indeed, Eqs 7-9 follow the idea from the Kingma et al.’s Semi-supervised VAE paper. The key difference is that their work assumed p_{\theta}(x,y,z)=p_{\theta}(x|y,z) p(y) p(z) where p(y) is the prior multinomial distribution, whereas we directly adopted p_{\theta}(x,y,z)=p_{\theta}(x|y,z) p_{\theta}(y|z) p(z) in our framework (shown in Figure 4). That is mainly because p_{\theta}(y|z) = p(y) might be a strong assumption. We will clarify these points in our revised paper.
>
> (4) Q: how to get Eq 4 from Lemme 4.2?
>
> A: Eq. (4) is derived directly from the definition of C-divergence in Eq. (2) (not from Lemma 4.2). As stated in Section 4.2, it is difficult to calculate the proposed C-divergence based on Eq. (2) as it uses the true underlying distributions in practice. So we propose to empirically estimate the C-divergence using finite examples across domains. In this case, the key derivation step from Eq. (2) to Eq. (4) is: Pr_{D}[{I(h), y=1} U {\Bar{I(h)}, y=0}] = Pr_{D}[h(x) = y] = 1 - Pr_{D}[h(x) != y] ~ 1 – 1/m \sum_{x,y} II[h(x) != y].
>
> (5) Q: if F-classifier always says that the example is from one domain?
>
> A: For Eq. (11) or the one above, it is impossible that the optimal F-classifier predicts all examples as one domain label. Note that C-divergence is defined on 1-min_F{…}. The inner minimization problem can be considered as a simple binary domain classification problem with at least accuracy 0.5 when we assume that the hypothesis class H is symmetric (i.e., 1 − h ∈ H if h ∈ H). Then in this case, even though there may exist an F-classifier predicting all examples as one domain label, it will not be optimal.
>
> (6) Q: relation to Continual learning?
>
> A: Thank you for pointing out this related work. We will include it in the next version. However, we would like to point out that our work is fundamentally different from Continual learning (such as Gradient Episodic Memory) in various aspects.
> (i) Different definition of negative transfer: in continual learning (such as Gradient Episodic Memory), negative (backward) transfer indicates that learning a new task decreases the performance on some preceding task (i.e., catastrophic forgetting); whereas we defined the negative transfer from the perspective of transfer learning across tasks, i.e.,  the prediction performance on the target task might be decreased using the knowledge from source task as compared to learning from the target task alone. Therefore, conventional continual learning aims to avoid catastrophic forgetting; in contrast, we focus more on studying the transferability across tasks.
> (ii) Different transferability measure: most of the existing continual learning algorithms follows the framework of pre-training on past tasks and fine-tuning on the new task. In contrast, we would like to explicitly minimize the domain discrepancy between the new task and past tasks for better transferability.

---

> > ### Author Response · Authors · 2020-11-17
> > **Response to AnonReviewer2**
> >
> > (7) Q: Typos?
> >
> > A: Thank you so much for pointing out the typos. We will revise it.
> >
> > (8) Q: what if a new version of the product released periodically with new sets of features?
> >
> > A: We would like to argue that our current distance measure and CONTE algorithm could be adapted to this problem setting. In this case, the distribution of the review data shifts in time. To be more specific, it would be biased due to sampling from different regions of the same target distribution. But in fact, we can also assume that the review data at each time stamp has its own data distribution (maybe slightly different from the true target distribution), and the individual data distribution of the reviews is evolving over time. Then, our algorithm can be easily adapted to this problem setting.
> >
> > (9) Q: Can we automatically learn \mu from the data?
> >
> > A: Our algorithm can be easily generalized with automatically learned \mu from the data. But this extension will introduce two additional challenges: (i) the overall model performance would depend on the weight estimation method; (ii) it has additional computational complexity on the weight estimation, which may be prohibitive for the dynamic setting. In fact, we have attempted to incorporate some existing weight estimation methods from multi-source domain adaptation algorithms (e.g., MDAN [1] and DARN [2]) into our CONTE algorithm. The results are given below on Office-31 (Webcam->DSLR). It is shown that the automatically learned weight values of \mu might lead to inferior predictive performance compared to the domain weight decay rate strategy. Recent research work [3] on multi-source domain adaptation has shown that simply assigning the weight ωj inverse proportional to domain discrepancy might not be optimal. This indicates that exactly estimating the weight of each historical task over the current task is much more challenging, and the sub-optimal solution is likely to lead to inferior performance. In contrast, the strategy with the domain weight decay rate enjoys better performance and high model efficiency.
> >
> >                             | T1 | T2 | T3 | T4 | T5
> > CONTE               | 0.9785 | 0.9235 | 0.9208 | 0.8886 | 0.9154
> >
> > CONTE_MDAN  |  0.9785 | 0.9356 | 0.8859 | 0.8027 | 0.7964
> >
> > CONTE_DARN   | 0.9785 | 0.9208 | 0.8644 | 0.7664 | 0.7463
> >
> > [1] Zhao, Han, Shanghang Zhang, Guanhang Wu, José MF Moura, Joao P. Costeira, and Geoffrey J. Gordon. "Adversarial multiple source domain adaptation." In Advances in neural information processing systems, pp. 8559-8570. 2018.
> >
> > [2] Wen, Junfeng, Russell Greiner, and Dale Schuurmans. "Domain Aggregation Networks for Multi-Source Domain Adaptation." In ICML, 2020.
> >
> > [3] Mansour, Yishay, Mehryar Mohri, Ananda Theertha Suresh, and Ke Wu. "A Theory of Multiple-Source Adaptation with Limited Target Labeled Data." arXiv preprint arXiv:2007.09762(2020).
> >
> > (10) Q: Does the information from Ti is retained in Tt from previous learning cycle, such that, it will be used by Tt+1 task?
> >
> > A: In our algorithm, the information from Ti would be retained in Tt because in the previous learning cycle, Ti would be used to learn the predictive function of T_t and then generate the pseudo-labels for unlabeled examples of T_t. Such information will be used by T_{t+1} task for measuring the C-divergence across tasks over feature and label information.

---

### Decision · Program_Chairs · 2021-01-07
**Final Decision**

**Decision:**

Reject

**Comment:**

The paper proposes transfer learning where the target domain data is evolving along time.  They use both labeled and unlabeled data to learn domain and time-invariant features based on a discrepancy measure they introduce.  Their proposed algorithm uses VAE to learn such features.  Reviewers have mixed response, although the author feedback did help.
The main limitation with the paper is that it does not seem to be aware of the very extensive literature on continuous domain adaptation.  The related work only discusses papers on transfer learning, multi-source domain adaptation, and continuous learning.  But ignores papers on continuous domain adaptation which are much more related to this paper.  The most recent of these that appeared in ICML 2020 also attempts to learn time invariant features using adversarial methods.  Unfortunately, the reviewers seem to be also unaware of this literature:

1.  Continuously Indexed Domain Adaptation,  Hao Wang, Hao He, Dina Katabi, ICML 2020
2.  Active Adversarial Domain Adaptation
3.  Continuous Domain Adaptation using Optimal Transport
4. Learning to Adapt to Evolving Domains - NeurIPS 2020
5.  Judy Hoffman, Trevor Darrell, and Kate Saenko. Continuous manifold based adaptation for evolving visual
domains. In Proceedings of the IEEE Conference on Computer Vision and Pattern Recognition, pages 867–874,
2014.
6 Massimiliano Mancini, Samuel Rota Bulo, Barbara Caputo, and Elisa Ricci. Adagraph: Unifying predictive and
continuous domain adaptation through graphs. In Proceedings of the IEEE Conference on Computer Vision and
Pattern Recognition, pages 6568–6577, 2019.

7 Atsutoshi Kumagai and Tomoharu Iwata. Learning future classifiers without additional data. In Thirtieth AAAI
Conference on Artificial Intelligence, 2016.